



# Observations of Sea Ice Melt from Operation IceBridge Imagery

Nicholas C. Wright[1], Chris M. Polashenski[1,2], Scott T. McMichael[3], Ross A. Beyer[3,4]

[1]Thayer School of Engineering, Dartmouth College, Hanover, NH, USA
[2]U.S. Army Cold Regions Research and Engineering Laboratories, Hanover, NH, USA
[3]NASA Ames Research Center, Moffet Field, CA, USA
[4]SETI Institute, Mountain View, CA, USA

*Corresponding author*: Nicholas Wright (ncwright.th@dartmouth.edu)

**Abstract.** The summer albedo of Arctic sea ice is heavily dependent on the fraction and color of melt ponds that form on the ice surface. This work presents a new dataset of sea ice surface fractions along Operation IceBridge (OIB) flight tracks derived from the Digital Mapping System optical imagery set. This dataset was created by deploying version 2 of the Open Source Sea-ice Processing (OSSP) algorithm to NASA's Advanced Supercomputing Pleiades System. These new surface fraction results are then analyzed to investigate the behavior of meltwater on first-year ice in comparison to multiyear ice. Observations herein show that first-year ice does not ubiquitously have a higher melt pond fraction than multiyear ice under the same forcing conditions, contrary to established knowledge in the sea ice community. We discover and document a larger possible spread of pond fractions on first year ice leading to both high and low pond coverage, in contrast to the uniform melt evolution that has been previously observed on multiyear ice floes. We also present a selection of optical images that captures both the typical and atypical ice types, as observed from the OIB dataset. We hope to demonstrate the power of this new dataset and to encourage future collaborative efforts to utilize the OIB data to explore the behavior of melt pond formation Arctic sea ice.





## 1 Introduction

The extent and age of the Arctic sea ice cover has declined since the beginning of the satellite record in 1979 (Stroeve et al., 2012). Ice melt is accelerated through albedo feedback cycles initiated by surface melt decreasing the ice cover's reflectance (Curry et al., 1995; Perovich et al., 2003). Understanding changes in sea ice properties that impact albedo, particularly melt pond coverage, is important to parameterizing sea ice in global climate models (Hunke et al., 2013; Serreze et al., 2009). In-situ observations that could support developing this understanding are sparse, difficult to acquire, and may not be broadly representative (Perovich, 2002; Wright and Polashenski, 2018). Remote sensing platforms provide a path to understanding sea ice surface change over larger scales. Newly developed computational techniques provide the means to analyze large remotely sensed datasets (Miao et al., 2015; Webster et al., 2015; Wright and Polashenski, 2018). The NASA Operation IceBridge project (OIB) has collected large amounts of high-resolution optical imagery of sea ice with the Digital Mapping System (DMS) (Dominguez, 2010, updated 2017). At ~10cm resolution, these images capture the ice surface in exquisite detail – but it is challenging to convert them to quantitative measures of ice conditions.

A new technique for analyzing high-resolution optical imagery of sea ice has recently been developed and demonstrated (Wright and Polashenski, 2018). This technique, named the Open Source Sea-ice Processing algorithm (OSSP), automatically analyzes input imagery and classifies image area into surface types such as melt ponds, unponded ice, and open ocean. Several improvements and new features that define version 2 of OSSP are presented here. This version was used to create a new dataset by deploying the algorithm on a large scale to process the entirety of the NASA OIB optical image dataset. This dataset is now publicly available for community use and for other studies leveraging the IceBridge data suite. This publication is intended partially to serve as supporting documentation for those uses.

The summer portion of the new dataset is then used to evaluate existing hypotheses about melt pond formation on Arctic sea ice. One such hypothesis describes the prevalence of ponds on first-year sea ice (FYI) versus multiyear ice (MYI). It has been widely hypothesized that FYI has a higher fractional pond coverage than MYI (Eicken et al., 2004; Fetterer and Untersteiner, 1998a; Morassutti and Ledrew, 1996; Perovich and Polashenski, 2012). This would contribute to positive ice-albedo feedbacks, since the higher pond fraction would lower albedo of FYI, re-enforcing the transition to a younger ice pack. The reasoning most cited for expecting higher pond coverage on FYI is related to ice and snow topography (Barber and Yackel, 1999; Derksen et al., 1997; Eicken et al., 2004). When sea ice grows in the relatively calm Arctic, it tends to form in flat, undeformed pans or fairly level pancake fields (Weeks, 2010). Though these pans are subsequently broken and ridged by dynamic forces, in most parts of the Arctic a large fraction of FYI remains level. When surface melt begins on level FYI floes, melt water is unconstrained by topography and spreads to cover a large fraction of the surface. On MYI, however, the ice has survived prior melt seasons. Prior melt creates more complex surface topography even in areas without mechanical deformation. The meltwater is then contained by the prior year's melt-formed topography into well-defined pools. The result *should be* that first-year ice would tend to experience greater pond coverage than multiyear ice. Indeed, this has been posited by several authors as a likely change in the Arctic (Eicken et al., 2004; Polashenski et al., 2012).



Despite this understanding, a collection of previous observations have shown the possibility that FYI may actually
have lower pond coverage than MYI under certain circumstances (Perovich, 2002; Webster et al., 2015). The new
observational dataset of melt ponds on sea ice from OIB is used here to test this more generally, revealing new
evidence that FYI *often* has lower melt pond fractions than neighboring multiyear ice.
A second, related, hypothesis on the behavior of FYI melt ponds suggests two summer melt evolution pathways
exist: one which yields high pond fraction, and one that yields near-zero pond fraction (Perovich, 2002; Polashenski
et al., 2017), depending on early season ice permeability and the duration of surface flooding. Our new observations
of pond coverage over large areas of FYI provide additional insight. Here, the OSSP-labeled OIB images were used
to assess the variation in pond coverage on FYI and the prevalence of pond-free floes within the Chukchi and Beaufort
Seas. To accomplish this, a method of post-processing has been developed that determines the size of sea ice areas
devoid of pond coverage as a metric to quantitatively address the prevalence of low pond coverage. This new analysis
reveals that FYI pond coverage indeed exhibits both pathways, but that there is not a strict duality – FYI pond coverage
appears to occupy all states across the near-zero to high coverage space.
**2 Methods**
**2.1 Data Sources**
The datasets described herein are the result of processing NASA Operation IceBridge optical DMS imagery. The DMS
images were acquired with a Canon EOS 5D Mark II digital camera which has a 10cm horizontal ground resolution
when used at the survey altitude of 1500 feet (Dominguez, 2010, updated 2017), and is available for download at the
National Snow and Ice Data Center (NSIDC). 87 IceBridge flights were processed, occurring between 2010 and 2018.
The OIB flights were categorized into freezing and melting conditions, which map to the spring/fall and summer
campaigns respectively. No flights took place during melt or freeze onset transitional phases, making this a clean
categorization: Flights before June 1st were categorized as freezing condition flights, and those taken after this date
were categorized as melting condition flights. There are several flights during fall freeze-up which are grouped with
the pre-June 1st images. Using this delineation, there were 8 flights during melting conditions and 79 flights during
freezing conditions. Of the 8 melting condition flights, 4 occurred in 2016 originating from Utqiaġvik, Alaska, and 4
occurred in 2017 originating from Thule AFB, Greenland. There was an additional summer flight departing from
Utqiaġvik on July 20th, 2016, that was not processed due to constant cloud cover obscuring the images.
A graphic of the flight tracks for all OIB sea ice flights processed, colored by freezing/melting condition status,
is presented in Fig. 1. For the majority of this paper, we will focus on the melting season (summer) flights, colored in
yellow. Spring data products are posted for use by the community. We anticipate that future analysis of spring flight
data will help confirm lead identification in analysis of altimetry data and provide statistics on lead size and spacing
and morphology useful to studies of, for example, blowing snow loss to leads or ice dynamics.


## 2.2 OSSP Algorithm Improvements

A number of improvements have been made to OSSP since the initial version 1 release described in Wright and Polashenski (2018). These changes can be divided into three categories: 1) Those that alter the algorithms used to classify images, 2) those which add new features, and 3) those which improve code efficiency but do not alter the core methodology. Changes that fall into category (3) reimplemented existing functions for improved performance and decreased computational resource usage. These will not be discussed in detail as they do not change the results.

### 2.2.1 Algorithm Refinements

OSSP is an object-based segmentation and classification image processing algorithm. In version 1, edge detection for segmentation was done by applying a Sobel-Feldman filter to the image, amplifying the resulting values to highlight strong edges, and thresholding low gradient value pixels to remove weak edges. The amplification factor and threshold value were both presented as tuning parameters that could control the number and strength of edges to detect in the image. In version 2, image edges are instead found with a Canny edge detector (van der Walt et al., 2014), which has three built-in tuning parameters: A gaussian filter with chosen radius that removes noise from the image, a high threshold which selects strong edges, and a low threshold which defines weak edges. These three parameters can be selected based on the quality of the input image and the degree of segmentation sought. The change in edge detection method does not significantly shift the behavior of the OSSP method but allows the user to better tune the segmentation to specific images. The remainder of the OSSP code uses methodology as presented in Wright and Polashenski (2018).

### 2.2.2 New Features

Four new features were added for processing the OIB optical image dataset: 1) An image quality analyzer which flags excessive cloud cover or haze, 2) an automatic white balance correction function, 3) expanded training datasets specific to OIB images, including shadow detection in spring images, and 4) orthorectification to a flat plane WGS84 spheroid.

Clouds and semi-opaque haze are common in OIB imagery. These often partly obscure the surface and prevent accurate image classification. An automated algorithm has been added that detects obscured images so that they can be removed from analysis. The quality check is based on applying a Fourier transformation to the image to detect the ratio of high and low frequency features. It is an implementation of the De and Masilamani (2013) method, where the quality score is the percent of image pixels that have a frequency greater than $1/100,000^{th}$ of the maximum frequency. Poor quality images were empirically found to have a score of less than 0.025, potentially unusable images had a score between 0.025 and 0.035, and images with a score greater than 0.035 were generally acceptable.

A large number of OIB images are taken in poor surface lighting conditions. This is often a result of the aircraft flying under cloud cover or high solar zenith angles. Darker than expected and blue-shifted images are observed under these conditions. Unlike the hazy images flagged by the quality check, these can still be accurately classified. An automatic white balance correction function has been added to standardize the hue and exposure of these images and the resulting image classification. We use a single-point white balance algorithm:





$$\begin{bmatrix} R_c \\ G_c \\ B_c \end{bmatrix} = \begin{bmatrix} \dfrac{omax}{R_w} & 0 & 0 \\ 0 & \dfrac{omax}{G_w} & 0 \\ 0 & 0 & \dfrac{omax}{B_w} \end{bmatrix} * \begin{bmatrix} R \\ G \\ B \end{bmatrix}$$

where
$$omax = \max(R_w, G_w, B_w)$$

and $(R_w, G_w, B_w)$ is a chosen white reference pixel, (R,G,B) is the original pixel value triplet, and $(R_c, G_c, B_c)$ is the
corrected pixel value triplet. The reference point triplet is chosen automatically based on the image histogram of each
color band; it is the smallest value that is both larger than the highest intensity peak and has less than 15% of that
peak's pixel counts. This method sets the selected reference point to true white (255,255,255). All other pixels in the
image are corrected with the same linear scaling which serves to both adjust the image exposure and rebalance the
RGB ratios. Limits to the scaling are in place to prevent images with a single surface from being improperly stretched
(e.g. an open water only image will remain black). The effect this has on two poorly illuminated images is shown in
Fig. 2.
The OIB dataset has a clear binary division between spring and summer flights. This characteristic allows for the
utilization of two specialized training datasets–one for each season. The summer training dataset is a new, larger, set
than was presented along with OSSP v1.0, including additional points to encompass a wider range of possible ice
conditions. The spring training dataset includes a ridge shadow surface classification class and does not include a melt
pond category. The shadow detection method was not applied to melting condition images as the typical summer solar
zenith angle yields fewer shadows. Removal of the melt pond category from spring images prevented occasional
spurious detection of melt ponds and improved the quality of results. The training data creation followed the same
technique presented in the OSSP version 1.0 documentation (Wright and Polashenski, 2018). The summer dataset was
expanded to a total of 1706 training points, and the spring dataset to a total of 865 points. These training datasets can
be found along with the OSSP code at (https://github.com/wrightni/ossp).
**2.3 Detecting Pond-Free Ice Areas**
The labeled image output by the OSSP algorithm was further analyzed to extract metrics about the spatial
distribution of water features in summer. First, the labeled image was converted into a binary image separating the
snow and ice features from water (i.e. melt ponds plus ocean). Next, the distance from every snow/ice pixel to the
nearest water feature was calculated, and peaks with a local maximum distance above a threshold of 15 meters were
recorded. Pond free areas are defined as a circle centered at these peaks with a radius of the distance to the nearest
water feature. Any two overlapping regions were combined by adding the non-overlapping area of the smaller region
to that of the larger region. These pond free regions are divided into two categories, small and large, based on a
threshold of a 27.5 m radius. The number of pond free areas per image was multiplied by the ice fraction (sum of all
non-ocean categories) of that image to account for differing ice concentrations between images. Figure 3 shows an
example of this detection, where the location of both the small and large regions are marked with small dots and the
large regions have a translucent circle showing the size of that region.





**2.4 Error**
There are several sources of error in OSSP ice type classifications when applied to the DMS dataset. The
established accuracy of the OSSP method, on a high-quality input image, is 96% (Wright and Polashenski, 2018). The
principle source of error novel to this OIB dataset was due to lower quality images, typically from haze obscuring the
surface or poor surface illumination. While automated methods standardize the quality of the input and flag bad images
(Section 2.2.2), some input errors remain. The impact of uncorrected haze is twofold: First, it causes the algorithm to
misclassify open water as melt pond, and second, it obscures surface type boundaries and causes insufficient image
segmentation. Both issues can be understood by looking at how haze changes an optical image: It adds noise to the
image, tends to brighten the pixel values, and blurs surface features. As the defining feature of open water is its uniform
darkness, a layer of haze makes this surface more like a dark melt pond. The blurring impacts the edge detection
algorithm used by OSSP and therefore causes a breakdown of the proper delineation of image surfaces. For the
analyses of the summer dataset presented herein, images were manually sifted to remove those scenes that were not
flagged by the QA analysis, but were still of questionable quality. Due to the heterogenous nature of sea ice, there is
a trade-off between accuracy on a specific image and accuracy on the entire dataset – some images flagged as low
quality may be usable with targeted processing. Users of this dataset should inspect their region of interest to ensure
the image quality meets their desired standard.
**3 Results**
**3.1 Melt pond fraction along OIB flight tracks**
In this paper we focus on presenting results from summer images only. Images from 87 IceBridge flights were
processed with the OSSP algorithm representing over 200,000 individual images using the methods described above
– these results are available for other investigations at the NSIDC archive. Figure 4 maps the track of every melt
season OIB flight and plots melt pond fraction observed along these tracks. Images where more than 70% of the area
was classified as open water are colored black. Images that were automatically removed due to a low quality score
(section 2.2.2) are colored orange, and images that were manually removed due to low source image quality are colored
red. The July 20[th], 2016 flight was not processed as there were not enough usable images in that flight.  Note both
high variation in pond coverage along track and general regional changes between flights. Some additional variation
between flights is due to temporal change, for example it appears a summer snow occurred just prior to the July 19,
2016 flight, lowering the observed pond fraction.
Figure 5 plots 300km of the along-track melt pond fraction for the July 24[th], 2017 flight. This figure illustrates
the large the variability possible in melt pond fraction along track seen in the first half of the flight (top), with a
minimum observed fraction of 10% and spikes to greater than 50%. The second half of this flight (bottom) has a more
uniform melt pond fraction of ~20%. Four peaks are highlighted in orange where a large blue pond formed on the
multiyear ice (See Fig. 11d). Figure 6a zooms in to a 10km subset of this transect, and the surface corresponding to
the orange highlighted section is shown in Fig. 6b. The optical image is the result of stitching 23 DMS images together.
The highlighted peak in melt pond fraction occurs on a section of first-year ice between two multiyear floes. This case





follows the prevailing hypothesis about the differences between pond formation on MYI and FYI. The relatively flat
FYI section allows melt ponds to spread over the surface more evenly, resulting in a higher melt pond coverage,
despite encountering the same atmospheric conditions as the MYI on either side. It is also possible that melt water
from the MYI drains to the lower elevation FYI (Fetterer and Untersteiner, 1998a).

**3.2 Influence of Ice Type on Melt Pond Fractions**

Each summer transect was categorized into first-year ice, multi-year ice, or mixed ice based on manual inspection of
those flight's images. The flights classed as a single ice type had at least 90% (estimated from visual inspection) of
that type. Melt pond statistics for single ice type flights are shown as box and whisker plots in Fig. 7, where each flight
is colored by its ice type categorization; blue for FYI and green for MYI. In these plots the box outline shows the 75th
and 25th percentile, the middle line displays the median, the whiskers show 1.5x the interquartile range, and the red
points are outliers. Generally, the 2016 flights departing from Utqiaġvik, Alaska, covered first-year ice while the 2017
flights departing from Thule AFB, Greenland, covered multiyear ice. There are three exceptions to this categorization:
July 13, 2016 and July 19, 2016 contain both ice types and flight A on July 25th, 2017 covers first-year ice. Statistics
for the two mixed ice type flights are plotted separately in Fig. 8, where each flight is divided into first-year or
multiyear ice categories.
Figure 7 reveals two insights into the difference in melt pond fractions between FYI and MYI. First, there is no
obvious difference in the median pond fraction between flights, and second, there is more variance in the pond
fractions on first-year ice. The variance is described by the interquartile range, the mean of which is 0.1 for the first-
year flights and 0.05 for the multiyear flights. In other words, while FYI exhibited a wider range of possible pond
fractions, the average coverage is not observed to be higher than on MYI.
However, this comparison may not address the hypothesis that pond coverage is higher on FYI because flight
lines occurring over two years and were exposed to unique forcing conditions. To investigate melt pond statistics
across ice that experienced similar forcing conditions, two flights that contained both FY and MY ice were selected
for further analysis: July 19, 2016 and July 13, 2016. The portions of these transects that depict each ice type were
manually determined. Results, delineated by ice type, for these two flights are shown in Fig. 8. The key observation
here is that the two flights show opposite relationships: On July 19 the FYI has a lower median pond fraction, while
on July 14, the MYI has a lower median pond fraction. These observations confirm FYI can exhibit a lower pond
fraction than multiyear ice under similar atmospheric forcing conditions. This suggests that pond evolution on FYI is
more variable than on MYI and demands we understand these apparently divergent evolutions.

**3.3 Observations of Pond-Free First-year Ice**

The frequency at which FYI develops very low pond coverage was investigated using the pond-free region detection
algorithm to find large unponded areas. Figure 9 shows the results of applying this algorithm to selected segments of
the July 19, 2016 flight. Panel (a) shows the results for a portion of primarily first-year ice with high pond coverage,
(b) shows a region of first-year ice that has many areas of pond-free ice, and (c) shows results from a section of
multiyear ice. The ice analyzed for Fig. 9a is what we understand would be considered as 'typical' first year ice by
most of the sea ice research community, with uniformly high pond fraction. This contrasts with the FYI analyzed for
Fig. 9b where, while melt ponds are still present, there are large open areas of pond free ice. The ponds on the MY
floe are regularly distributed and the fractional pond coverage shows little variance. Expanding from these regions of
this specific flight, 17% of all summer FYI images processed for this study have 3 or more large pond free regions. In
contrast, in the MYI portion of this dataset, only 5% of images have 3 or more large pond free regions. While there is
a clear difference between the MYI and FYI types, the important observation here is the large percentage of our FYI
images that exhibit pond behavior different from the assumed standard of high coverage.

**3.4 Snapshots of a Summer Sea Ice Cover**

In processing the Operation IceBridge optical imagery dataset, we have had the unique opportunity to review a
significant library of images detailing different sea ice states, looking at thousands of square km of sea ice. So few
people actually observe the sea ice that notions of what is 'typical' or unusual are still not well known. In this section
we present some examples of what we have observed to be 'representative' ice states, and examples of ice conditions
that are uncommon. These are intended to serve as a qualitative summary of the extensive OIB observations, against
which future campaigns can be quickly compared. For each presented image we label the noted features based on the
frequency at which we have observed them. Along an arbitrary 100km transect of ice in a given melt state; *common*
describes a feature that occurs on most or all of the ice, *occasional* describes features that would be expected to show
up 5-10 times, and *infrequent* describes a feature that may present once or twice.
Sea ice scenes shown in Fig. 10: **(a)** First year ice that shows a wide range of the possible melt pond fractions,
ranging from pond free to high pond coverage; *occasional*. **(b)** Highly ponded level first year ice scene in early melt,
where ice appear as islands in a sea of water. Such ice was *common* in large areas in the Chukchi sea. **(c)** First year
ice with high pond fraction and very interconnected pond structure. *Common*; this represents the generally understood
behavior of first-year ice. Here we also see that ponds preferentially form towards the middle of the floe leaving a
pond-free border around the edge. The floe-edge gradients are particularly strong in this image, the pond-free border
is an *occasional* feature. **(d)** Example of a floe where ponds preferentially form away from the edges. These small
floes with central ponds were common in broken first-year ice. **(e)** First year ice in the Lincoln sea. Ponds have started
to drain already, as evidenced by the drainage channels visible throughout the ice. This type of relatively low coverage
and consolidated ponds were *infrequent* in the OIB dataset. We speculate that deep snow dunes and thick ice are
responsible. **(f)** This image shows a region that appears to have had a recent summer snowfall event. The snow serves
to fill shallow ponds with slush or to completely cover them and significantly lowers pond fraction – *infrequent* as it
is dependent on specific weather conditions. **(g)** A *common* example of high pond fraction first-year ice. Note that this
scene includes some sediment laden ice, which is also *common*. **(h)** Flat and thin ice pans that are almost completely
covered by melt water, this scene is *common* for late stages of melt on FYI.
Sea ice scenes shown in Fig. 11: **(a-c)** *Common* examples of ponded multiyear ice floes with characteristically
blue ponds that are well consolidated by surface topography, showing the range of pond fractions that are possible.
**(d)** Example of large reservoir-like ponds that were only observed on multiyear ice. These are *occasional* features on
large sections of multiyear ice. **(e)** Multiyear ice with first-year inclusions from ocean that refroze during the last





winter, this is *common* for multiyear ice at lower latitudes, and *occasional* at higher latitudes. In cases of small FYI
inclusions in MYI fields like this, the FYI ice is typically darker had has a higher pond coverage. **(f)** An example of
low pond coverage MYI – this was *infrequent* in the OIB dataset. **(g+h)** Ponded first year ice undergoing drainage,
where evidence of previous ponds is still visible. The overall image represents *common* features, but the drainage
pattern here is *infrequently* observed, likely due to its short lifespan.
**4 Discussion**
**4.1 Variation in Pond Coverage on FYI Precludes Simple Relationship with MYI**
A general consensus in the sea ice community indicates that FYI has, on average, higher melt pond coverage than
does MYI. While such an understanding of ponds is not universally held, it is prevalent and represents a testable
hypothesis which our results above did not support. The reasoning for the hypothesis is two part, covering both early
season FYI ponding (when meltwater sits on impermeable ice above sea level) and late season FYI ponding (after
ponds have drained to sea level). In the early season case, it is argued that with limited topography, a similar volume
of meltwater will flood larger areas of FYI than it would cover on rougher MYI. This is supported by observations in
early melt stages, which show FYI melt pond coverage in excess of 60%. Such coverage exceeds that seen on multiyear
ice at any time (Landy et al., 2014; Polashenski et al., 2012). In the late season case, it has been argued that thinner
FYI will have less buoyancy and less ice area above freeboard. In both cases, FYI ponds would be greater than MYI.
An alternate hypothesis about the behavior of FYI ponds emerging in some recent papers is that FYI pond
coverage is extremely variable and may have bimodal evolution driven by snow topography and permeability
(Polashenski et al., 2017; Popović et al., 2018). FYI ponds may not form at all under certain circumstances if the ice
is highly permeable or lacks snow cover (Polashenski et al., (2017) and references therein). Other observations show
very high melt pond coverage that persists even after ponds drain to sea level (Polashenski et al., 2015). These
divergent possibilities of pond behavior raise the possibility of bimodal behavior wherein some FYI would flood
extensively and experience more ponding than MYI while other FYI might not pond at all. The prevalence of these
two very different types of behavior would be key to understanding whether the transition from MY to FYI is
increasing pond prevalence. No large scale, comprehensive observations have been available to resolve how prevalent
such behaviors are.
Our image dataset provides some such information on the nature of FYI ponding. The time covered by the images
is late in the melt season, when FYI is fully permeable and ponds, if any formed, have drained to sea level (see
Polashenski et al., (2012) for a description of the stages of pond evolution). Evidence of pond drainage features is
common, and we conclude the ponds are largely at sea level. Polashenski et al., (2012) showed that ponds remaining
after pond levels drain to sea level are simply those areas where the ice surface floats below sea level. The divergent
pond behavior is then topographically forced. If the surface of the ice is level when ponds drain, the ice surface will
be uniformly above sea level, leaving pond-free ice. If, however, snow dunes or differential melt creates roughness on
the surface, some of the surface will protrude from the ocean significantly and other areas will not, creating the
possibility for ponds to remain at sea level. If subtle topography is powerful, we expect FYI pond coverage late in the



year would be highly variable, likely low on FYI that remains smooth, higher on moderately rough FYI, and lower
again on the roughest FYI (see Popović et al., (2018) for more discussion). Given the range of outcomes and range of
snow/ice topography on FYI, there would not be a characteristic relationship between pond coverage on FYI and
adjacent MYI.
Examining the pond coverage in more detail provides evidence that the range of possible melt states is larger on
first-year ice than it is on multiyear ice. In other words, FYI exhibits all possible states between low and high coverage,
while MYI pond fraction typically exists within a small window. Returning to the boxplots in Fig. 7, note the larger
interquartile range (IQR) of the first-year flights versus the multiyear flights. If we were to accept the traditional
hypothesis that all first-year ice had high pond cover, we would expect the FYI to have a higher median but a similar
IQR. However, this is not the case. These observations suggest pond cover on FYI is highly variable, and only in a
subset of circumstances does the ice exhibit the expected higher pond fraction. Examples of each behavior are included
in Fig. 8. The traditional understanding of melt pond evolution on FYI, where flat undeformed ice allows melt water
to spread horizontally and create large areas of pond covered ice is often observed on landfast ice or ice attached to a
multiyear floe (e.g. Barber and Yackel, 1999; Derksen et al., 1997; Fetterer and Untersteiner, 1998b; Uttal et al.,
2002). For example, Fig. 6b shows a refrozen lead between two multiyear ice floes, where the pond fraction is
significantly higher on the flat FYI than on either of the adjoining MYI floes. Freely floating floes of flat FYI often
exhibit little to no pond cover late in the melt season (as seen in Fig. 10d). We also note many examples of floes that
are pond free along their edges, such as in Fig. 10c, and floes that exhibit nearly complete pond coverage (such as
10b,g,h). This dataset, therefore, helps establish that no simple relationship between FYI and MYI ponding exists, and
that the transition to FYI is not causing uniformly higher melt pond fraction, as has been expected. The highly variable
nature of FYI ponding is, however, regionally coherent, strongly suggesting that the history of conditions the ice is
subject to governs ponding. Connecting conditions to pond prevalence is therefore a topic worthy of investigation for
better understanding FYI albedo feedbacks.
**5 Conclusion**
A new dataset quantifying sea ice surface fractions observed in Operation IceBridge DMS imagery has been created
using the recently developed OSSP algorithm. This dataset classifies the surface coverage into four categories. During
the melt season these categories are: 1) snow or thick ice, 2) dark or thin ice, 3) melt ponds and submerged ice, and 4)
open water. In freezing conditions, the categories become 1) snow or thick ice, 2) dark or thin ice, 3) open water, and
4) ridge shadows. The dataset allows for the investigation of sea ice surface type distributions along OIB transects and
opens the door for new studies, both by analysing this dataset in isolation (as demonstrated here), and by combining
it with coincident OIB datasets such as ice thickness or ice roughness. This dataset will be available at the NSDIC for
community use. Future improvements to this dataset should include work towards a more sophisticated haze removal
algorithm to apply to the OIB optical images. This will increase accuracy and increase the fraction of images that can
be successfully processed.
We have investigated a common hypothesis regarding the characteristics of melt pond development on FYI vs
MYI and discovered evidence that it may be unsupported. FYI does not necessarily develop larger melt pond fractions



than multiyear ice, even under the same atmospheric forcing conditions. We have presented additional evidence that first-year sea ice exhibits much larger variance its evolution; where there is not one path that defines the typical behaviour of pond coverage. We suggest future process studies investigate the mechanisms that drive FYI towards high or low pond fraction and specifically note that time-series image observations and/or field studies may be necessary to unravel this question. The different trajectories that pond development can apparently take on FYI may have large impacts on sea ice modelling efforts, through albedo feedbacks. Furthermore, we suggest combining this new melt pond dataset with data available from the IceBridge Airborne Topographical Mapper to determine the relationship between sea ice topography and melt pond formation.

*Data and Code Availability.* The OSSP algorithm code is available on github (https://github.com/wrightni/ossp) and the release for this manuscript is archived at zenodo (DOI: 10.5281/zenodo.3551033). The pond free detection algorithm will be archived at zenodo prior to publication and is available at github (https://github.com/wrightni/pondfree_detection) during review. Raw Operation IceBridge DMS imagery is available from the National Snow and Ice Data Center (https://doi.org/10.5067/OZ6VNOPMPRJ0). OSSP generated results are being uploaded and archived at the NSDIC and will be available prior to publication of this manuscript.

*Author Contributions.* NW was responsible for writing the original draft, creating the data visualizations, review and editing of the manuscript, designing and testing the OSSP software, conceptualization and programming of the pond-free detection algorithm, and formal analysis of the OSSP generated results. CP was responsible for funding and supervision for the Dartmouth/CRREL team, project administration, writing and editing the manuscript, and consulting on methodology and result analysis. SM was responsible for implementing the OSSP software on NASA's Pleiades system, monitoring data processing, and data archiving. RB was responsible for funding acquisition and supervision for the Ames Research Center team and for review and editing of the manuscript draft.

*Conflicts of Interest.* The authors declare that they have no conflicts of interest.

*Acknowledgements*. The authors would like to thank NASA's AIST Program whose funding enabled this research. The image processing for this work was carried out on NASA's Advanced Supercomputing Pleiades system.



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





**Figures**

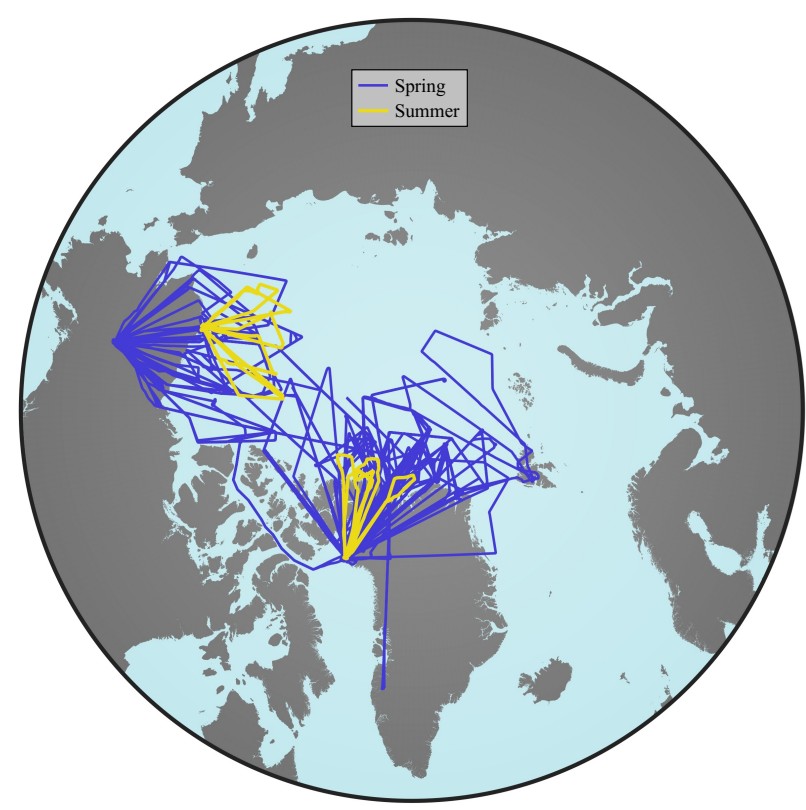

**Figure 1. Plot of all flights processed with OSSP, colored by the melt conditions during the flight. Spring freezing conditions in blue, and summer melting conditions in yellow.**



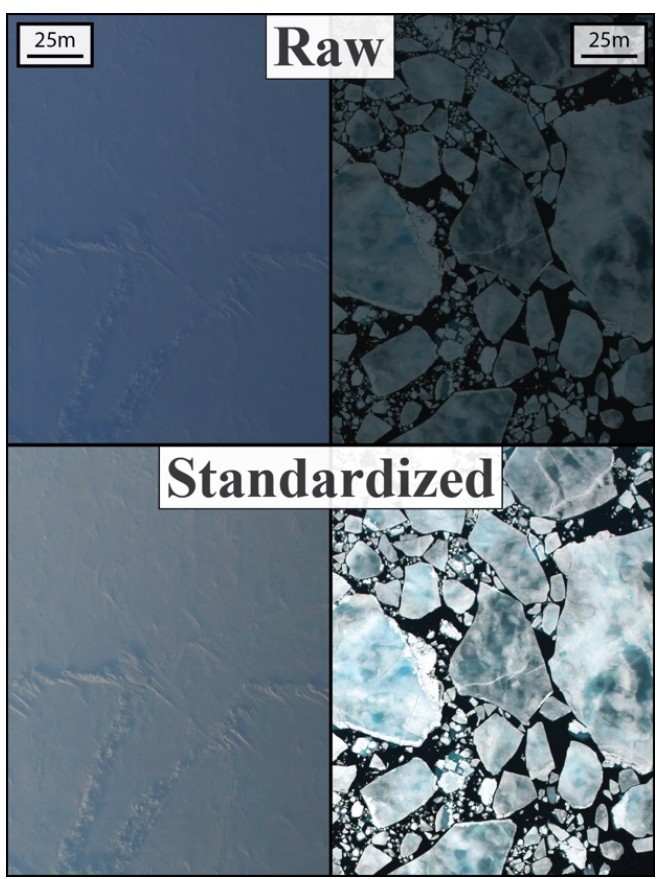

**Figure 2. Demonstration of the image preprocessing steps. The raw images (top) have poor surface illumination and a blue hue, both of which have been removed in the standardized images (bottom).**



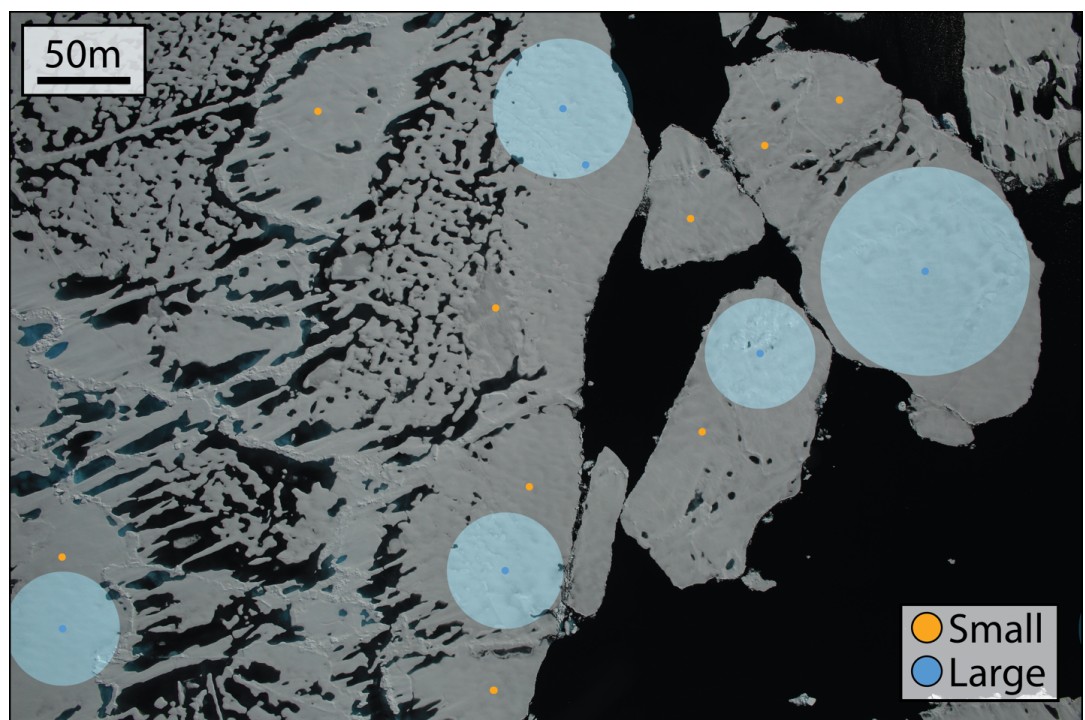

**Figure 3. Example of the pond free region detection. Pond free regions are marked by small colored dots, where blue dots indicate the larger regions and orange indicates the smaller ones. Translucent blue circles are drawn with a radius equal to the size of the detected large regions. Blue dots without a translucent circle were merged with a neighboring region.**





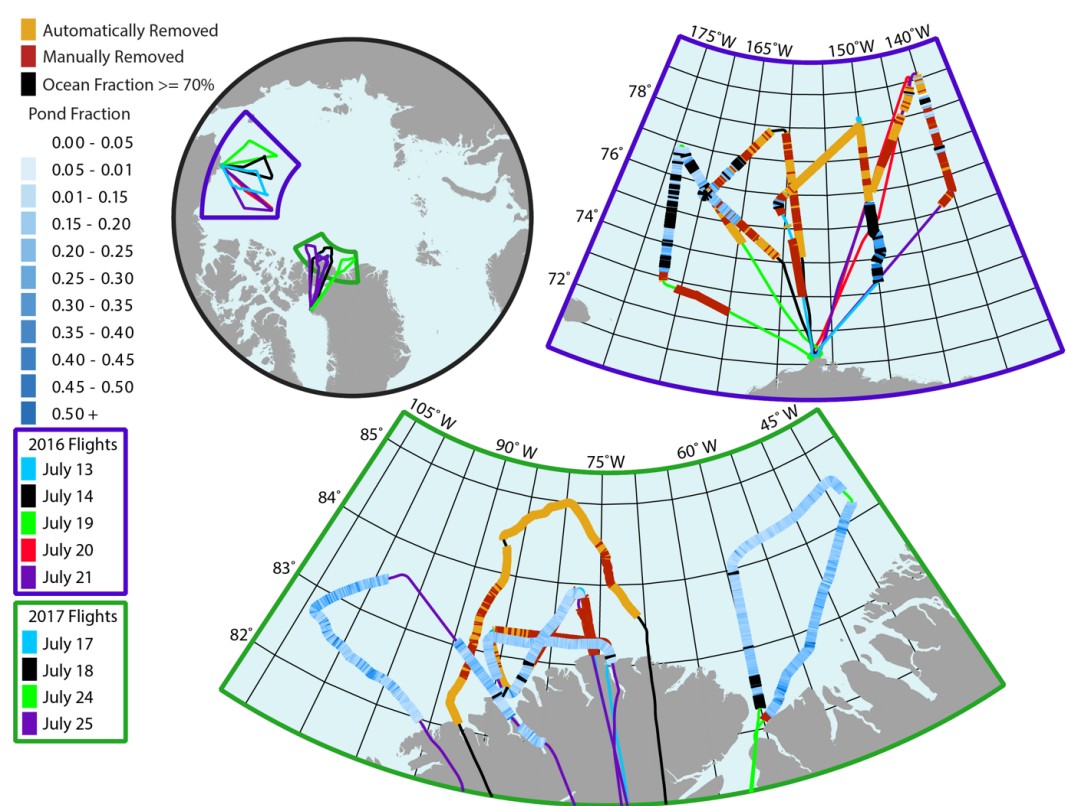


**Figure 4. Melt pond fraction along OIB summer transects. Automatically and manually removed images are indicated by orange and red, respectively. 2016 flights were more prone to haze obscuring the ice surface and therefore have more deleted images.**



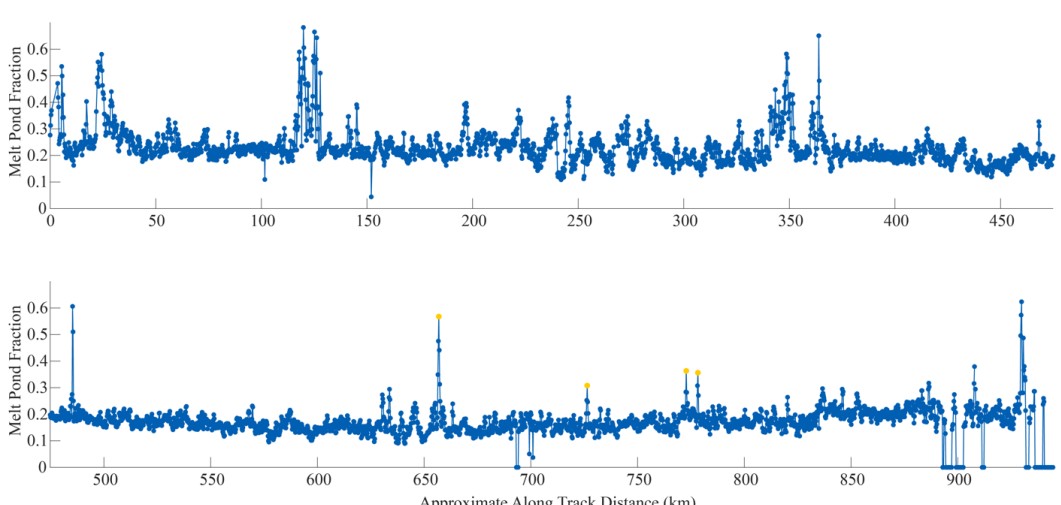

**Figure 5. Melt pond fraction along track for flight July 24, 2017. The four orange highlighted points represent areas where there was a large blue pond on the multiyear ice that occupied a large fraction of the image. See Fig. 11d for an example of this feature.**

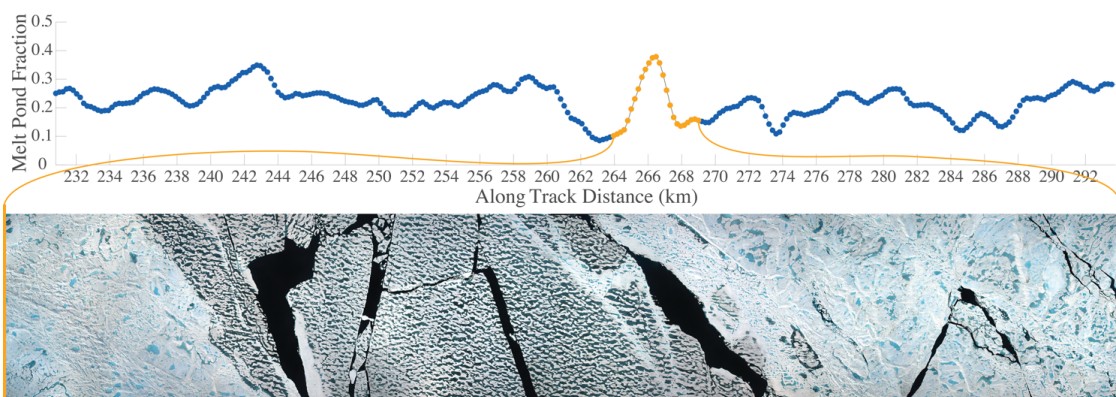

**Figure 6. Melt pond fraction along a several kilometer section of the July 24, 2017 flight. The orange highlighted region is depicted as a series of stitched together DMS images that show a first-year inclusion between two multiyear floes.**





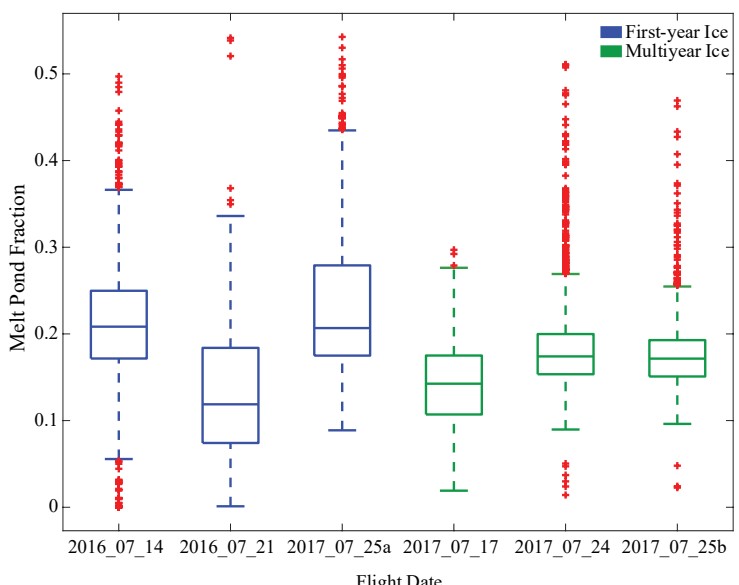


**Figure 7. Melt pond statistics from summer OIB flight which contained only a single ice type. Blue corresponds to first year
ice statistics, green to multiyear ice statistics, and red crosses indicate outliers.**




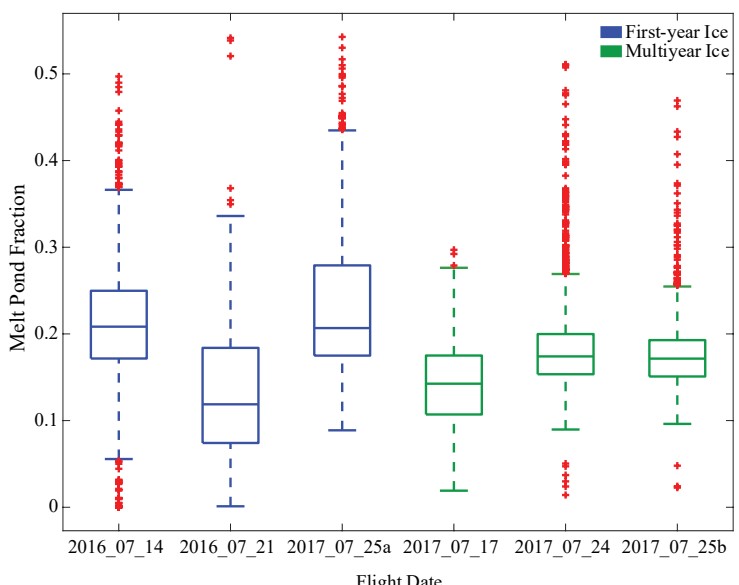


**Figure 8. Melt pond statistics from two flights that contain both first-year and multiyear ice. In the July 13 case, multiyear
ice has a lower pond fraction, while in the July 19 case the first-year ice has a lower pond fraction. Blue corresponds to first
year ice statistics, green to multiyear ice statistics, and red crosses indicate outliers.**







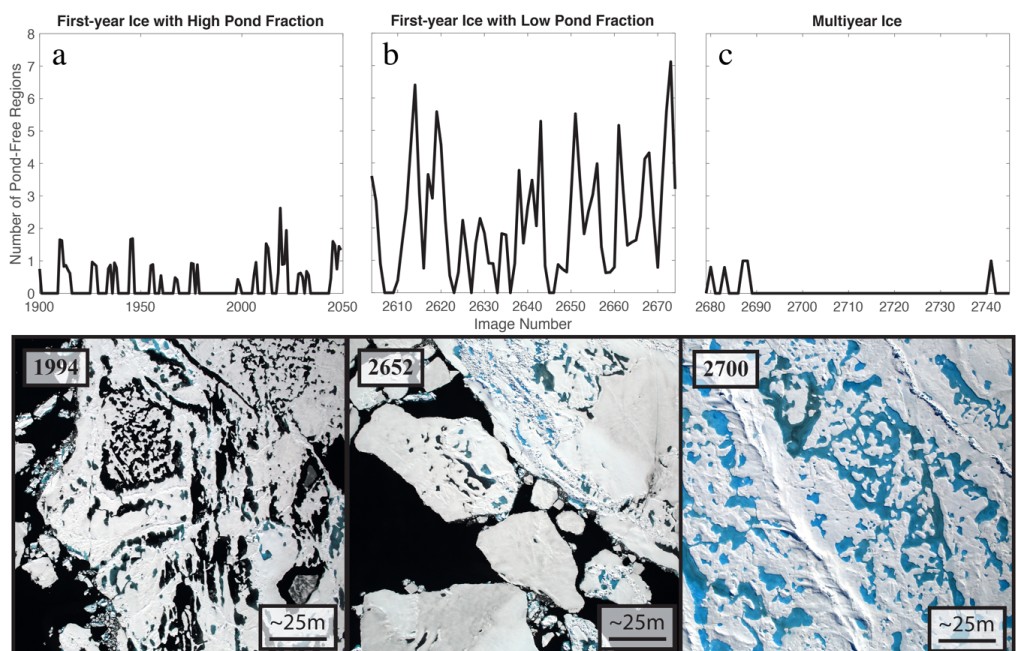

**Figure 9. Number of pond-free regions on several regions of sea ice observed during the July 19, 2016 flight (top) and a sample image representing that region of ice (bottom).**





474

**Figure 10. Exhibits of sea ice surface features as seen in the DMS dataset. Each panel is a full IceBridge image, and while flight altitude affects image resolution, each scene is approximately 600 m by 400 m. See text for full description of each frame.**



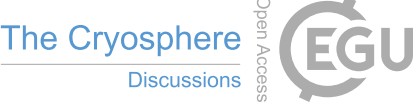

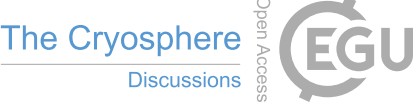

478

**Figure 11. Exhibits of sea ice surface features as seen in the DMS dataset. Each panel is a full IceBridge image, and while flight altitude affects image resolution, each scene is approximately 600 m by 400 m. See text for full description of each frame.**