# Peer review of "Observations of Sea Ice Melt from Operation IceBridge Imagery"

_The Cryosphere, 2019_

## Referee Comment (RC1) · Anonymous Referee #1 · 20 Jan 2020

The paper describes an analysis of surface conditions of Arctic sea ice in summer. The images are processed from a previously developed algorithm, which is approved upon and well-described, and all methods and output are made publicly available. The paper is very well-written, logically organized, and the figures are illustrated clearly. More details need filling in for parts of the methods, which should be straight-forward to address. The largest concerns I have are the testing of the hypothesis that melt pond coverage over first-year ice is higher than that of multiyear ice and the two pathways of melt pond evolution suggested for first-year ice. These concerns can be remedied by reconsidering the argument and taking into account the following points:

(1) Melt pond evolution is variant in nature, particularly over first-year sea ice. Operation IceBridge sampled melt ponds at different stages of melt given the long distances

covered. To assume all pond formation and evolution progressed the same, for example assuming all ponds had drained (as in the discussion), is a stretch even within the same survey line. By sampling over such large, regional areas, these surveys are sampling different states of melt pond evolution. (2) The bimodal pathway argument of pond evolution for FYI is a gross oversimplification. While it is an interesting idea to consider, the argument that FYI is either pond-free or heavily ponded during summer is weakly supported. Melt pond coverage on first-year ice ranges from no ponds to heavily-ponded with everything in between based on available data.

Please see the following suggestions for further improvements: L49. Relatively calm Arctic. Calm relative to what? The Arctic seas are dynamic.

L57-60. The introduction would benefit from more description about melt pond evolution. One aspect that's missing is the transitory coverage of melt ponds with melt. At one stage, FYI melt ponds may have lower coverage than MYI melt ponds. At a later stage of melt, the same FYI melt ponds may have greater coverage than MYI melt ponds. Pond coverage can change substantially depending on the ice state and progression of melt.

L67-68. I recommend tweaking the language here. While the results do show low and high coverage of melt ponds on FYI, which is a valuable finding, the results do not directly link together melt pond coverage and the processes posed in Polashenski et al. 2017.

L77. No flights took place during melt or freeze onset transitional phases. How was this determined?

L116-117. Were there specific cases of high-quality imagery discarded using this method? It's worth mentioning in the text in case there are any biases worth considering.

L131-132. Are these limits subjective to each image or is a standard value applied to

all? How were the limits determined?

L134. Is a clear, binary division true for flights where freezing and recent snowfall took place?

L137-140. Is there an option for using melt pond and shadow detection in the algorithm on late spring or early summer images when both conditions are present? It would be worth noting this in text here.

L144/Section 2.3. What new information does the number of pond-free areas provide that the areal ice fraction doesn't? It would be helpful to discuss this in a sentence or two here. For one, the distribution of pond-free ice has implications for disparate surface melt rates and the new pond-free metric would seemingly give more information in this respect.

L148/L152. 15 m and 27.5 m values are specific. How were they chosen?

L170. What is meant by targeted processing?

L176. What are the results exactly? Are they segmented images or simply surface fractions of all images? Please clarify here.

L177. Please define melt pond fraction. Is it the areal fraction of the image scene or of the sea ice? How are melted-through ponds within an ice floe classified?

L177-178. Why were images with 70% ocean area discarded? Melt pond fractions in these images would be useful information.

L179. What is meant by low source image quality? Does this mean that there were images that had low light, were hazy, that the automation didn't catch before? If so, it would be helpful to state how many images (the fraction of the total) the automation removed. This can tell us how much work the automation saves us from doing and approximately how much work is left to do using this method.

L180. Not enough to do what? Do the authors mean that there was no usable imagery

from that flight?

L189-190. It would be useful to see the equivalent segmented image of 6c as an additional panel to the figure.

L196-198. What's the error associated with the ice type classification? How was second-year ice classified?

L214-215. How were the ice types distributed along the surveys? Were FYI and MYI well-mixed or was one ice type located predominantly north, east, etc.? It'd be helpful to note their distribution here.

L217-219. The first sentence needs more description. Work by Eicken et al. 2002 and Webster et al. 2015 demonstrated the same result, but what this analysis shows is that it can happen on a regional scale rather than a local scale, and that's important. The second sentence can also be expanded on. Several previous studies showed pond evolution between FYI and MYI differ. What's new with this study is the link to the large-scale variability in pond coverage. For example, one could hypothesize that there should be less spatial variability in MYI pond coverage on a regional scale because it's less variable in time relative to pond evolution on FYI. These results support that hypothesis.

L231-232. This sentence is unclear.

L241. How was "most" defined? Was this 51% of the ice area or more than 10 times?

L243/Figure 10. This is a nice result. I was hoping to see the equivalent segmented image. It'd be worthwhile to include this either in the main text or as supplementary information.

L250-253. Is this shorefast ice? It's worth stating so if it is, as it may be typical for shorefast FYI in this region.

L254-255. I'd suggest rephrasing this to "infrequent" to the OIB observations, since it
may be a common phenomenon.

L256. It would be helpful to circle or highlight the sediment-laden ice as it's not apparent in this image. It also raises the question, does the algorithm also detect sediment-laden ice or is it detected as a melt pond?

L258/Figure 11. Similar comment as Figure 10, it'd be helpful to see the segmented equivalent in the main text or supplementary information.

L267-319/Section 4.1. Please see main concerns above.

L280-281. The lack of ponds in Polashenski et al. 2017 seemed to be due to a snowfall event and freezing conditions rather than high permeability and a lack of snow.

L284-287. Do the results from earlier works using MODIS data not apply here?

L289. How was high permeability and pond drainage determined on such a large spatial scale? Figure 10b shows no drainage features. This surface condition was classified as common in the dataset, which conflicts with the next sentence.

L293-294. Is this what's being suggested for the pond-free FYI areas? Before, the argument was that ponds never formed?

L296. It's not clear what is meant by if subtle topography is powerful.

L298-300. This is not clear.

L312-313. Is this statement in reference to the OIB data set? For previous works, this was not found to be the same. It would be worth clarifying here.

L322-324. This description should be described near the beginning of the manuscript. Submerged ice may contribute to a larger proportion of pond fraction for FYI than MYI.

L333-335. Similar to the main concerns above, a snapshot of lower FYI pond coverage than MYI pond coverage does not address the hypothesis. Previous works have shown pond coverage on FYI to be highly temporally variable over summer compared to that

on MYI. The temporal average of melt pond fraction for FYI and MYI over the melt period may indeed support the hypothesis.

Figure 4. It would be helpful to use a more dynamic color scheme for the melt pond fraction. It's difficult to see the distribution along the survey lines.

Figure 8. It would be helpful to know the sample size for each case.

[Figure]

---

## Short Comment (SC1) · 11 Feb 2020

Major comments:

(1) A number of questions remain about the algorithm performance and the error analysis could be strengthened:

L156. Other than haze, what are the main sources of object misclassifications? L156. How do object misclassifications impact the derived melt pond fraction? On Line 137, you state the shadow detection method is not applied because "typical summer solar zenith angle yields fewer shadows." The sun angle is still low in the Arctic and ridge shadows do exist in the summer. How are shadows that do exist in the imagery classified if they do not have their own category? Are they classified as melt pond? How

does excluding this step impact results? How does aircraft attitude and altitude impact the impact pixels and hence, the classification algorithm and derived melt pond fraction? Have the authors re-quantified the algorithm error, given the modifications to the algorithm (Section starting at Line 107), since Wright and Polashenski, 2018?

(2) Designation of ice type

The authors state on L203 that the flight on July 25th 2017 covers first year sea ice. This does not seem justifiable for two reasons. a) the authors provide their own definition of a FYI flight (Line 197, that 90 % of the images in the flight are FYI). Given this definition, and visual inspection of the DMS imagery from the flight, it is not obvious that the flight is over predominantly FYI. A larger percentage of images with pressure ridges and rubbled ice, indicating a long deformation history, and thus, MYI. Many images resemble the MYI depicted in Figure 11a-c and described as "common examples of ponded multiyear ice floes with characteristically blue ponds that are well consolidated by surface topography" (Line 258). b) the location of the flight line north of Ellesmere Island in the Central Arctic is over sea ice known to be the oldest and thickest ice in the Arctic, and highly unlikely to be predominantly FYI in origin. The 2017 Arctic Report Card found that the ice in this region in March is predominantly MYI (Figure 3c, Perovich et al., 2017). Given that it is well know that the ice in this region is some of the thickest ice in the Arctic (e.g. Figure 2b, Sallila et al. 2019), this area is highly unlikely to be predominantly FYI.

For reference: Perovich, D., Meier, W., Tschudi, M., Farrell, S., Hendricks, S., Gerland, S., Haas, C., Krumpen, T., Polashenski, C., Ricker, R., & Webster, M. (2017). Sea Ice [in Arctic Report Card 2017], http://arctic.noaa.gov/Report-Card Sallila, H., Farrell, S. L., McCurry, J., & Rinne, E. (2019). Assessment of contemporary satellite sea ice thickness products for Arctic sea ice. Cryosphere, 13(4).

(3) Forcing conditions affecting sea ice floes in survey area

L212. "To investigate melt pond statistics across ice that experienced similar forcing

conditions, two flights that contained both FY and MY ice were selected for further analysis" How do the authors know this ice has experienced similar forcing conditions throughout its lifetime? Considering the Beaufort Gyre is known to be an especially dynamic area, the ice observed during the flight surveys may have come from different regions. The ice in this region may, at the time of the survey, be experiencing uniform forcing conditions, but the assumption that all ice covered in a survey has experienced similar forcing conditions throughout its lifetime is invalid.

(4) Melt pond fraction calculation clarification:

L175. How is melt pond fraction calculated? If the OSSP algorithm classifies melt ponds and submerged ice in the same category, is submerged ice included in the melt pond fraction calculation? How does the inclusion of submerged ice impact the melt pond fraction parameter?

L177. Why do the authors choose images with open water area < 70% as a threshold for displaying melt pond fraction results? Do you include images with open water area > 70% in melt pond fraction results (Section 3.2 and 3.3).

Minor questions needing clarification in the text:

L80 Are data collected on 15 July 2016 analyzed? This flight survey is plotted in Figure 1, but no results are shown (Figure 4).

L180. The authors state that data from the 20 July 2016 flight were not processed because "not enough usable images" How do the authors determine what was enough?

L184. Does Figure 5 follow Figure 4, and only show melt pond fraction for images with open water area < 70%?

L203: Can you distinguish between the 25July2017 flight A and flight B within the text and/or in Figure 4 (where they are currently shown in the same color)?

L323. How is a melt pond defined in this study? Is a melt pond still a melt pond when it

has melted through the sea ice? What about other features: melting snow, thaw holes, algae on ice?

Figure 4. Bottom figure. For the 17 July 2017 and 18 July 2017 flights, it looks like there are no images remaining for analysis. Is that correct? Can you provide the total number of images analyzed for each flight, and total discarded? Perhaps this information could be included in a table or added to the figure.
* * *

---

## Referee Comment (RC2) · Anonymous Referee #2 · 27 Feb 2020

Summary

The authors provide an update to the Open Source Sea-ice Processing (OSSP) algorithm and apply it to the optical Digital Mapping System (DMS) images acquired during Operation Ice Bridge flight tracks flown in melting conditions. The OSSP derived relative surface fractions include ice, open water, and melt pond. Statistics on melt pond fraction are important for understanding sea ice evolution, light exchange, and for parameterizing models. The documented improvements to the OSSP are important since the code is being made freely available and potentially facilitates some standardization in the processing of high resolution optical datasets of sea ice during melting conditions.

In general the paper is well written and organized, and the output figures and tables

concise and informative. The improvements to the OSSP are well documented, however there are some problems with the analysis of the output data from the OSSP applied to the optical DMS data from the Ice Bridge flights. The assertion that, based on the analyzed data, first-year ice (FYI) often has lower melt pond fraction than multiyear ice (MYI) is misleading. There is insufficient data analyzed, and the temporal component of melt pond fraction evolution (including a comprehensive review by one of the co-authors) is mentioned but largely ignored for the purpose of supporting the assertion. Lines 288-293 describe the timing of the acquisition of the DMS images for this study as being in late in the melt season, when ponds have drained to sea level. In this case it can be expected that, for any sea ice that is still above sea level, the mechanically weak FYI will have likely drained and melt pond fraction will be lower than it is for MYI undergoing similar melting conditions. That is consistent with the stage of melting, not the overall behavior of FYI and MYI during melting conditions. The hypotheses in the introduction are therefore poorly stated, the analysis misguiding, and the resulting conclusions are flawed. That FYI experiences greater melt pond fraction than MYI has been more than posited, as stated on line 55, it has been well studied in the context of sea ice geophysical evolution. The authors must analyze their data in the context of the fairly well understood temporal behavior of melt pond fraction evolution on FYI and MYI, and situate their observations in the correct context (late season), using ancillary data if needed. It would make more sense to present the data as is, and evaluate the OSSP algorithm performance, without the general assertions about FYI and MYI behaviors – this not detract from some very interesting results.

Other comments 1. In cases where the sea ice has melted to sea level, and the ice floats below sea level, that is ocean water and sea ice – not melt pond covered sea ice. Has this been correctly specified in the algorithm and resulting statistics?

2. Consistent terminology regarding the season and stage of melt would make the paper clearer and easier to follow. For example, are spring conditions (line 86) actually spring when it is freezing conditions? The June 1st cut-off for categorizing freezingmelting conditions is arbitrary.

3. More information on the nature of the training data is required. It would be interesting if the algorithm could be trained to detect drained FYI (i.e. ice previously covered by pond which has then drained once connectivity with the ocean is achieved), since this ice has much different fluid and gas exchange properties compared to exposed ice.

4. Once FYI and MYI are defined the full terms are not required.

5. The assertion on line 225 is biased. Consideration of typical melt pond fraction conditions would include temporal domain, not just the spatial. This has been well documented. There could very well be low pond fraction if the FYI has drained and I would suggest that the sea ice community is aware of this.

Detailed Comments

L=Line L32: 'fine' detail instead of exquisite

L73-74: specify the extent i.e. ground coverage of the images

L108-109: more detail on expanded training datasets is needed

L145: Start this section by defining a pond-free ice area. Otherwise it is a bit confusing, as all areas of exposed ice (1-PF) are pond-free ice areas.

L185: "...the large the variability ..." delete extra 'the'

L217-219: There has been much work done understanding the melt pond fraction evolution for FYI and MYI, and pond evolution is likely explained by drainage mechanisms in this late period.

L269-277: Missing from this paragraph is the occurrence of late season FYI when ponds have drained but the ice is still above sea level. In this case, FYI pond fraction would be less than MYI (likely the case in Figure 10f, for example).

L282-285: There should be mention of diurnal variations in pond fraction due to variable meltwater input and drainage process which, for level sea ice, can lead to dramatic changes in melt pond fraction over very short periods of time. Subtle changes in air temperature or surface energy balance can predicate these changes in melt pond fraction.

L331-332: This hypothesis is not investigated in the paper since it does not utilize data from early stages of melt pond coverage, when ice is relatively impermeable and differences in melt pond fraction are related to topography hence ice type.

L443: The blue color scheme for pond fraction is difficult to interpret in the figure.

---

## Author Comment (AC1) · 8 May 2020

*Author's replies are denoted with a > before the paragraph. PDF with track changes included as supplement.

The paper describes an analysis of surface conditions of Arctic sea ice in summer. The images are processed from a previously developed algorithm, which is approved upon and well-described, and all methods and output are made publicly available. The paper is very well-written, logically organized, and the figures are illustrated clearly. More details need filling in for parts of the methods, which should be straight-forward to address. The largest concerns I have are the testing of the hypothesis that melt pond coverage over first-year ice is higher than that of multiyear ice and the two pathways of

melt pond evolution suggested for first-year ice. These concerns can be remedied by reconsidering the argument and taking into account the following points:

>Thank you for your review of our manuscript. You have provided great insights that have helped to improve this work. We have made efforts in the revised document to better discuss the time dependent nature of melt evolution, and to reassess how we can investigate these hypotheses with the snapshots provided by our dataset.

Melt pond evolution is variant in nature, particularly over first-year sea ice. Operation IceBridge sampled melt ponds at different stages of melt given the long covered. To assume all pond formation and evolution progressed the same, for ex- ample assuming all ponds had drained (as in the discussion), is a stretch even within the same survey line. By sampling over such large, regional areas, these surveys are sampling different states of melt pond evolution.

>This is true and may actually be a point in favor of assessing regional mean melt pond fraction. While the flight lines are temporally static, their long spatial footprint means, as you point out, that we are sampling ice in many different states of pond evolution. Given that the data samples a range of ice states the fact that we did not observe any statistically significant difference in the mean melt pond fraction between ice types is suggestive. We concede that this investigation alone is not sufficient to prove or disprove the hypothesis that FYI has higher MPF than MYI, but believe these observations are an important addition to that discussion.

The bimodal pathway argument of pond evolution for FYI is a gross oversimplification. While it is an interesting idea to consider, the argument that FYI is either pond-free or heavily ponded during summer is weakly supported. Melt pond coverage on first-year ice ranges from no ponds to heavily-ponded with everything in between based on available data.

>We agree with your assessment that the bimodal pathway is an oversimplification. Early looks at this dataset led credence to this hypothesis, so we set out to test it more
formally. Our results here show that the bimodal pathway is not supported, as you point out, and our intent is to show this. We have therefore reworked several sections throughout the manuscript to be clearer on this point. We want to convey the idea that FYI is more variable than MYI, exhibiting all states from low to high coverage, but that there is not a bimodal path as initially posited.

>Note the last sentence of the introduction: "This new analysis reveals that FYI pond coverage indeed exhibits both pathways, but that there is not a strict duality – FYI pond coverage appears to occupy all states across the near-zero to high coverage space."

Please see the following suggestions for further improvements: L49. Relatively calm Arctic. Calm relative to what? The Arctic seas are dynamic.

>Changed this sentence to be clearer. Here we are trying to establish the well observed predominance of flat topography on FYI and not get lost in the details about ice growth mechanisms.

L57-60. The introduction would benefit from more description about melt pond evolution. One aspect that's missing is the transitory coverage of melt ponds with melt. At one stage, FYI melt ponds may have lower coverage than MYI melt ponds. At a later stage of melt, the same FYI melt ponds may have greater coverage than MYI melt ponds. Pond coverage can change substantially depending on the ice state and progression of melt.

>We have added a discussion on the four stages of melt pond evolution. We have also added additional details to this section that looks at previous author's evidence for FYI with low pond coverage.

L67-68. I recommend tweaking the language here. While the results do show low and high coverage of melt ponds on FYI, which is a valuable finding, the results do not directly link together melt pond coverage and the processes posed in Polashenski et al. 2017.
>We have added this qualification to the introduction: "While the OIB image dataset provides large spatial coverage over long flight transects, the lack of temporal coverage makes it impossible to directly link these snapshots of pond coverage to any specific pond evolution process."

L77. No flights took place during melt or freeze onset transitional phases. How was this determined?

>This categorization was determined from established knowledge on when melt onset and melt pond formation typically begins. The only summer flights were in late July, well into melting conditions everywhere in the Arctic during 2016 and 17. We have added additional details referencing passive microwave derived melt onset dates to help with this categorization.

L116-117. Were there specific cases of high-quality imagery discarded using this method? It's worth mentioning in the text in case there are any biases worth considering.

>We did not encounter this issue. This flagging system is conservative and is more likely to not flag problematic images than it is to flag good ones. This is the reason for having to supplement the flagging with manual inspection.

L131-132. Are these limits subjective to each image or is a standard value applied all? How were the limits determined?

>These limits are standard, but only applied to select images. The limit for the white reference value is 200. These limits are only applied to images that do not contain both ice and ocean, which is determined by the number of peaks in the intensity histogram and the dynamic range of the image (the difference between the darkest peak and the brightest peak). We have added these details to the text.

L134. Is a clear, binary division true for flights where freezing and recent snowfall took place?

>Yes. While we agree there is much variability in sea ice conditions – specifically that periodic freezing and snowfall events often occur in summer months – our intent here is to separate the obviously different ice conditions between March/April (prior to melt pond formation) and those of late July (after melt pond formation). As this division is solely because of melt pond detection, we feel comfortable separating the flights into "expect melt ponds to be present" and "expect no melt ponds".

L137-140. Is there an option for using melt pond and shadow detection in the algorithm on late spring or early summer images when both conditions are present? It would be worth noting this in text here.

>The algorithm allows for this, but this was not done for the dataset described here. A new training dataset could be produced to incorporate both melt and shadow surface classifications, or even other surfaces entirely. We have added some text explaining this flexibility of the OSSP code.

L144/Section 2.3. What new information does the number of pond-free areas provide that the areal ice fraction doesn't? It would be helpful to discuss this in a sentence or two here. For one, the distribution of pond-free ice has implications for disparate surface melt rates and the new pond-free metric would seemingly give more information in this respect.

>You are exactly right, the primary benefit here is the information it provides on the spatial distribution of melt ponds. We see a difference in this metric between certain types of FYI and MYI even if the total MPF is the equivalent between them. This is because on MYI the ponds are evenly distributed across the surface (few pond-free zones) while on FYI the ponds can be clumped in areas of high pond fraction with other regions pond-free. This metric also provides insight on different types of FYI – FYI that has many pond free areas is experiencing some difference in melt evolution than FYI with well distributed ponds.

L148/L152. 15 m and 27.5 m values are specific. How were they chosen?

>These values are misleadingly specific but were chosen to be roughly 2x and 4x the mean caliper diameter of melt ponds. We have changed the values to be 12m and 25m for clarity, rerun the analysis (results were the same) and added our justification for the threshold to the text and a citation for the mean caliper diameter value.

L170. What is meant by targeted processing?

>Here we meant tailoring a training dataset to process a specific subset of images, rather than one that performs well across a large variety of input images. We have changed the wording here to make this clearer.

L176. What are the results exactly? Are they segmented images or simply surface fractions of all images? Please clarify here.

>The results are classified images – where each image pixel has been given a value based on its classified state. These can then be readily converted into simple surface fraction numbers.

L177. Please define melt pond fraction. Is it the areal fraction of the image scene or of the sea ice? How are melted-through ponds within an ice floe classified?

>MPF is a fraction of the ice area, not image area. We have added a sentence here clarifying how melt pond fraction is determined. Melted through ponds are classified as open water following from the arguments in Wright and Polashenski, 2018. In short – we approach this from a solar radiation energy balance perspective where melted through ponds are more similar to ocean in their radiative properties. Submerged ice is classified as "melt pond" for the same reason.

L177-178. Why were images with 70% ocean area discarded? Melt pond fractions in these images would be useful information.

>A single IceBridge image typically only covers 600x400m. If 70% of this is ocean, then melt pond fractions calculated from this small area are very easily skewed by large ponds (this area is well below the "aggregate scale"). Note that the images are

still processed, we just don't show the pond fraction in this plot. Even full images have a small enough area for the melt pond fraction to be skewed by large melt ponds, as shown by the orange dots in Figure 5.

L179. What is meant by low source image quality? Does this mean that there were images that had low light, were hazy, that the automation didn't catch before? If so, it would be helpful to state how many images (the fraction of the total) the automation removed. This can tell us how much work the automation saves us from doing and approximately how much work is left to do using this method.

>Yes – the manually removed images were ones with clouds/haze that were not detected by the automated system. We have added the percent caught automatically versus manually.

L180. Not enough to do what? Do the authors mean that there was usable imagery from that flight?

>There were not enough clear images to justify the effort needed to process and filter the results. A statistically relevant sample would not have been created with the small number of usable images.

L189-190. It would be useful to see the equivalent segmented image of 6c as an additional panel to the figure.

>We assume here you mean the final labelled image? Image segmentation is a specific term to describe an intermediate step of our algorithm. We have added a many of the images presented in the text as classified images in a supplemental figure.

L196-198. What's the error associated with the ice type classification? How was second-year ice classified?

>Second year ice would fall into the multiyear ice category, though it depends on the estimated surface roughness. These delineations are visually based, so the separation is between flat and undeformed ice versus textured and aged ice rather than a definitive

knowledge of the ice age.

L214-215. How were the ice types distributed along the surveys? Were FYI and MYI well-mixed or was one ice type located predominantly north, east, etc.? It'd be helpful to note their distribution here.

>For flights that observed both ice types in the Beaufort/Chukchi regions there were pockets of MYI in the northern regions of a predominantly FYI pack. Otherwise the flights were only a single ice type (using our >90% estimation). We have included this information in the text.

L217-219. The first sentence needs more description. Work by Eicken et al. 2002 and Webster et al. 2015 demonstrated the same result, but what this analysis shows is that it can happen on a regional scale rather than a local scale, and that's important. The second sentence can also be expanded on. Several previous studies showed pond evolution between FYI and MYI differ. What's new with this study is the link to the large-scale variability in pond coverage. For example, one could hypothesize that there should be less spatial variability in MYI pond coverage on a regional scale because it's less variable in time relative to pond evolution on FYI. These results support that hypothesis.

>These are good insights, and we have reworked this section to better reflect what is new in this study and what has been previously observed. We have also added additional content to the discussion section to better address these concerns.

L231-232. This sentence is unclear.

>We have rewritten this sentence.

L241. How was "most" defined? Was this 51% of the ice area or more than 10 times?

>Changed to be "... that can be expected on more than half of the ice".

L243/Figure 10. This is a nice result. I was hoping to see the equivalent segmented

image. It'd be worthwhile to include this either in the main text or as supplementary information.

>We have included this as a supplemental figure.

L250-253. Is this shorefast ice? It's worth stating so if it is, as it may be typical for shorefast FYI in this region.

>Yes, this ice is likely shorefast ice north of Ellesmere Island. We have changed this description: "(e) Shorefast level ice in the Lincoln sea. Ponds have started to drain already, as evidenced by the drainage channels visible throughout the ice. This type of relatively low coverage and consolidated ponds were infrequent in the OIB dataset, but may be common of ice in this region"

L254-255. I'd suggest rephrasing this to "infrequent" to the OIB observations, since it may be a common phenomenon.

>This is likely true, and we have added this extra information.

L256. It would be helpful to circle or highlight the sediment-laden ice as it's not apparent in this image. It also raises the question, does the algorithm also detect sediment-laden ice or is it detected as a melt pond?

>This image is actually not a great example of sediment laden ice, so we have removed this description from the text. Sediment-laden ice does not have its own classification category and would likely be put into the gray ice category, or possibly melt pond, depending on its color and darkness.

L258/Figure 11. Similar comment as Figure 10, it'd be helpful to see the segmented equivalent in the main text or supplementary information.

>We have included this as a supplemental figure.

L267-319/Section 4.1. Please see main concerns above.

>Revised discussion section, see comments in response to main concerns.

L280-281. The lack of ponds in Polashenski et al. 2017 seemed to be due to a snowfall event and freezing conditions rather than high permeability and a lack of snow.

>Polashenski et al., (2017) also discusses observations of pond-free ice that appears to have never had a snow cover (Specifically in reference to the satellite image in their Figure 15). We have added a citation to Eicken et al., 2004 here, which discussed the relationship of snow cover to pond formation.

L284-287. Do the results from earlier works using MODIS data not apply here?

>It is the authors opinion, supported by our own recent study (Wright and Polashenski, 2020), that existing MODIS melt pond products do not have the accuracy required to answer this question.

L289. How was high permeability and pond drainage determined on such a large spatial scale? Figure 10b shows no drainage features. This surface condition was classified as common in the dataset, which conflicts with the next sentence.

>If we look at the OIB dataset as a whole, the majority of the observed surface is in an advanced state of melt where the ponds have drained to sea level. This was determined empirically from looking at the dataset. This surface condition is common in reference to ice that is in a similar state of melt. In 10b, the state of melt can be described as ice that has not yet drained to sea level.

L293-294. Is this what's being suggested for the pond-free FYI areas? Before, the argument was that ponds never formed?

>We think that both pathways are possible. If the ice does not have the snow cover to support ponding (as noted by Eicken et al., 2004), or if ice permeability is too high to allow ponding (when the ice warms before surface melt begins the pore space cannot refreeze when freshwater enters, meaning ponds cannot form above sea level (Polashenski et al., 2017)), then the ponds will never form. In this section we are dis-

cussing the mechanisms required for pond free ice to emerge from ice that did have initial ponding.

L296. It's not clear what is meant by if subtle topography is powerful.

>We have removed this phrase and revised this section.

L298-300. This is not clear.

>This section has been reworked for clarity.

L312-313. Is this statement in reference to the OIB data set? For previous works, this was not found to be the same. It would be worth clarifying here.

>This statement is in reference to the OIB dataset, and we have clarified this here.

L322-324. This description should be described near the beginning of the manuscript. Submerged ice may contribute to a larger proportion of pond fraction for FYI than MYI.

>We have added the official category descriptions to the introduction of this manuscript.

L333-335. Similar to the main concerns above, a snapshot of lower FYI pond coverage than MYI pond coverage does not address the hypothesis. Previous works have shown pond coverage on FYI to be highly temporally variable over summer compared to that on MYI. The temporal average of melt pond fraction for FYI and MYI over the melt period may indeed support the hypothesis.

>We have revised the conclusion section to be clearer about the conclusions that we can and cannot draw from our dataset. As you pointed out, some of our claims were too bold to address with temporal snapshot datasets.

Figure 4. It would be helpful to use a more dynamic color scheme for the melt pond fraction. It's difficult to see the distribution along the survey lines.

>We have increased the contrast for this figure.

Figure 8. It would be helpful to know the sample size for each case.

>This has been added.  

 

Please also note the supplement to this comment:
https://www.the-cryosphere-discuss.net/tc-2019-288/tc-2019-288-AC1-
supplement.pdf
* * *
Ice is white, ponds are blue, and ocean is black.

**Fig. 1.** Supplement figure 1, classified versions of Figure 10. Ice is white, ponds are blue, and ocean is black.

**Fig. 2.** Supplement figure 2, classified versions of Figure 11. Ice is white, ponds are blue, and ocean is black.

---

## Author Comment (AC2) · 8 May 2020

\* Author responses are denoted with a > before the paragraph. A PDF with changes tracked is included as a supplement.

Summary The authors provide an update to the Open Source Sea-ice Processing (OSSP) algorithm and apply it to the optical Digital Mapping System (DMS) images acquired during Operation Ice Bridge flight tracks flown in melting conditions. The OSSP derived relative surface fractions include ice, open water, and melt pond. Statistics on melt pond fraction are important for understanding sea ice evolution, light exchange, and for parameterizing models. The documented improvements to the OSSP are important since the code is being made freely available and potentially facilitates some

standardization in the processing of high resolution optical datasets of sea ice during melting conditions.

In general the paper is well written and organized, and the output figures and tables concise and informative. The improvements to the OSSP are well documented, however there are some problems with the analysis of the output data from the OSSP applied to the optical DMS data from the Ice Bridge flights.

>Thank you for your review of our manuscript. You have provided some insightful comments that have helped us to improve this work. In general, we have made a number of changes to better include discussions of the temporal aspects to melt pond formation and to properly place our observations in the context of known pond evolution pathways. We have attempted to remove or lessen the more speculative discussion points in the original manuscript and better incorporated previous research that supports our analyses.

The assertion that, based on the analyzed data, first-year ice (FYI) often has lower melt pond fraction than multiyear ice (MYI) is misleading. There is insufficient data analyzed, and the temporal component of melt pond fraction evolution (including a comprehensive review by one of the co-authors) is mentioned but largely ignored for the purpose of supporting the assertion.

>We do not believe this is misleading. FYI and MYI have unique pond evolution, and it is expected that FYI will fall below MYI during certain phases of melt. According to the four stages of melt documented by Eicken et al. (2002), this happens during stage two, where FYI drains much faster than MYI. We have, however, removed the phrase "often" as we do not have the data to support this for the whole season. Observations at SHEBA found 10-30% of FYI to have zero or low pond coverage late in the melt season (Eicken et al., 2004), and our results (17%) fit right in the middle of this range.

Lines 288-293 describe the timing of the acquisition of the DMS images for this study as being in late in the melt season, when ponds have drained to sea level. In this case it can be expected that, for any sea ice that is still above sea level, the mechanically weak FYI will have likely drained and melt pond fraction will be lower than it is for MYI undergoing similar melting conditions. That is consistent with the stage of melting, not the overall behavior of FYI and MYI during melting conditions.

>At late stages in FYI pond evolution, any sea ice that is still above sea level is by definition unponded because no ponds exist above freeboard. We may be considering different definitions of a melt pond than you because we are approaching this from an albedo and radiative transfer perspective, where submerged ice falls into the melt pond category. This is consistent with prior research where on FYI the "melt pond fraction" steadily increases in stage 3 after FYI ponds have become fully connected with the ocean (Eicken et al., 2004, Polashenski et al., 2012).

>For illustration we have included a pair of images below. Panel A of shows FYI in an advanced state of melt that can be assumed to be thin ice with ponds that are fully connected to the ocean water, yet the surface is almost entirely flooded. Contrast this with Panel B, which was taken the same day just a few km away, where the FYI has very little pond cover.

The hypotheses in the introduction are therefore poorly stated, the analysis misguiding, and the resulting conclusions are flawed.

>The hypotheses are both presented in similar form in previous work (as cited in the manuscript). We have, however, attempted to make it clearer in our manuscript that our statement of these hypotheses in the introduction does not mean that we have confirmed them to be true. Quite the opposite! For example, we did not see sufficient evidence for the duality hypothesis and rejected it (as much as it is possible to do so with this dataset).

That FYI experiences greater melt pond fraction than MYI has been more than posited, as stated on line 55, it has been well studied in the context of sea ice geophysical evolution. The authors must analyze their data in the context of the fairly well understood temporal behavior of melt pond fraction evolution on FYI and MYI, and situate their observations in the correct context (late season), using ancillary data if needed. It would make more sense to present the data as is, and evaluate the OSSP algorithm performance, without the general assertions about FYI and MYI behaviors – this not detract from some very interesting results.

>We have made a number of refinements to better include discussions on the temporal aspects of melt pond evolution and remove assertions that are not sufficiently supported by the temporal snapshots provided with this dataset.

Other comments 1. In cases where the sea ice has melted to sea level, and the ice floats below sea level, that is ocean water and sea ice – not melt pond covered sea ice. Has this been correctly specified in the algorithm and resulting statistics? Consistent terminology regarding the season and stage of melt would make the paper clearer and easier to follow. For example, are spring conditions (line 86) actually spring when it is freezing conditions? The June 1st cut-off for categorizing freezing-melting conditions is arbitrary.

>Submerged ice is classified with the melt pond category following from the arguments in Wright and Polashenski, 2018. In short – we approach this from a solar radiation energy balance perspective where submerged ice is more similar to a melt pond in its radiative properties. Melted through ponds are classified as open water for the same reason. We have added these categorizations to the introduction and the terminology through the paper is consistent with these definitions.

>The June 1st cut off is arbitrary but is not important for the categorization. We have changed the description of the cutoff to be in reference to mean melt onset date from passive microwave datasets.

More information on the nature of the training data is required. It would be interesting if the algorithm could be trained to detect drained FYI (i.e. ice previously covered by pond which has then drained once connectivity with the ocean is achieved), since this

ice has much different fluid and gas exchange properties compared to exposed ice.

>More detailed information on the training data is available in Wright and Polashenski, 2018, where this method was first presented. The training datasets here are larger but are the same in other regards as those previously described.

>The ability to detect drained FYI would be powerful but it is likely not possible from optical datasets. Drained ice in many cases does not look different than melting ice that never had a pond cover.

Once FYI and MYI are defined the full terms are not required.

>We have replaced the full terms with the abbreviations after the first use.

The assertion on line 225 is biased. Consideration of typical melt pond fraction conditions would include temporal domain, not just the spatial. This has been well documented. There could very well be low pond fraction if the FYI has drained and I would suggest that the sea ice community is aware of this.

>Bias implies some ulterior motive or misrepresentation to support a goal, which is not our intention. We agree that specifying the 'typical' melt pond cover on FYI depends on the temporal domain because the pond fraction evolves over the melt season and have therefore clarified our statement here.

>We have changed the phrasing in this section to include mention of the temporal aspect of pond formation. Our goal is to point out the prevalence of pond free ice observed in our dataset and to place this in context with previous studies, not to claim that pond free ice is a novel observation. Because our dataset is a snapshot in time we cannot determine if the pond free ice was the result of pond drainage or the result of ice that never formed ponds.

>We have also included references to previous work that have observed pond free ice.

Detailed Comments

L32: 'fine' detail instead of exquisite

>Changed.

L73-74: specify the extent i.e. ground coverage of the images

>Added this information.

L108-109: more detail on expanded training datasets is needed

>More detail is available in the publication that describes this technique. There is not much else to add beyond what is in that manuscript.

L145: Start this section by defining a pond-free ice area. Otherwise it is a bit confusing, as all areas of exposed ice (1-PF) are pond-free ice areas.

>We have moved the definition to the beginning of this section.

L185: ". . .the large the variability . . ." delete extra 'the'

>Fixed.

L217-219: There has been much work done understanding the melt pond fraction evolution for FYI and MYI, and pond evolution is likely explained by drainage mechanisms in this late period.

>We have reworked this section to include more discussion of previous work and to place it into the context of known MPF evolution for FYI and MYI.

>Drainage is a possible explanation, but there is also the possibility that ponds never formed on this ice. We cannot investigate that from this dataset because there is no temporal dimension.

L269-277: Missing from this paragraph is the occurrence of late season FYI when ponds have drained but the ice is still above sea level. In this case, FYI pond fraction would be less than MYI (likely the case in Figure 10f, for example).

>We have added a few sentences here discussing times where FYI would be expected to have lower MPF based on previous studies:

" These effects must be balanced with the times in melt evolution where FYI is expected to have lower MPF. In the early season, MPF on FYI tends drop faster than on MYI because the meltwater is able to drain to sea level at a faster rate (Polashenski et al., 2012), and in the late season thicker FYI pond fractions would be lower than MYI because the more of the level surface sits above freeboard (e.g. Figure 10d). "

L282-285: There should be mention of diurnal variations in pond fraction due to variable meltwater input and drainage process which, for level sea ice, can lead to dramatic changes in melt pond fraction over very short periods of time. Subtle changes in air temperature or surface energy balance can predicate these changes in melt pond fraction.

>We have added discussion of diurnal effects on melt pond fraction to this paragraph.

L331-332: This hypothesis is not investigated in the paper since it does not utilize data from early stages of melt pond coverage, when ice is relatively impermeable and differences in melt pond fraction are related to topography hence ice type.

>We have rewritten this paragraph of the conclusions to fix this issue:

>"We have investigated snapshots of melt pond coverage differences between FYI and MYI in the Beaufort/Chukchi Sea region for 2016 and the Lincoln Sea for 2017. Our results support previous findings by X and Y that FYI can have lower pond fraction than MYI under the similar forcing conditions. While the results presented herein cannot definitively confirm or refute the hypothesis that FYI has higher mean pond fraction than MYI, the high variability in FYI pond fraction over large regions suggests that the general rule of thumb that FYI should have higher ponding than MYI is too simplistic. Furthermore, the finding that FYI exhibits much larger variance its evolution indicates that there is not one path that defines the typical evolution of pond coverage. We did not find sufficient evidence that there is a strict duality in FYI pond evolution either, and we suggest future process studies investigate the mechanisms that drive FYI towards high or low pond fraction and [. . .] "

L443: The blue color scheme for pond fraction is difficult to interpret in the figure.

>We have adjusted the contrast in this figure.

Please also note the supplement to this comment: https://www.the-cryosphere-discuss.net/tc-2019-288/tc-2019-288-AC2-supplement.pdf
* * *
[Figure]

**Fig. 1.** Example of pond free and high pond cover ice in close spatial proximity.

**Supplement:**

[revised manuscript text omitted]

Supplemental Figures:

[Figure]

Figure S1. Classified versions of the images shown in Figure 10. White regions are snow/ice, blue regions
are melt ponds are submerged ice, and black regions are open water.

[Figure]

Figure S2. Classified versions of the images shown in Figure 11. White regions are snow/ice, blue regions
are melt ponds are submerged ice, black regions are open water.

---

## Author Comment (AC3) · 8 May 2020

Author responses are denoted by a > before the paragraph.

Major comments: A number of questions remain about the algorithm performance and the error analysis could be strengthened:

L156. Other than haze, what are the main sources of object misclassifications?

>Haze is the main source of bulk object misclassification. Transitional surfaces (e.g. dark melt ponds, very thin ice) are the second highest source of misclassification. However, because these surfaces are typically transitioning between categories it is difficult to determine their "true" category in the first place. These and other sources are discussed in more detail in the error analysis section of the document describing

the OSSP methods (Wright and Polashenski, 2018).

L156. How do object misclassifications impact the derived melt pond fraction? On Line 137, you state the shadow detection method is not applied because "typical summer solar zenith angle yields fewer shadows." The sun angle is still low in the Arctic and ridge shadows do exist in the summer. How are shadows that do exist in the imagery classified if they do not have their own category? Are they classified as melt pond? How does excluding this step impact results? How does aircraft attitude and altitude impact the impact pixels and hence, the classification algorithm and derived melt pond fraction? Have the authors re-quantified the algorithm error, given the modifications to the algorithm (Section starting at Line 107), since Wright and Polashenski, 2018?

>Previous work has determined that in spring imagery ridge shadows make up less than 0.5% of the total ice area (Webster et al., 2015) and are therefore a small source of error even if always misclassified. Their impact would be expected to be even lower in summer, where the sun angles are higher. Misclassified shadows are typically assigned a label of melt pond, and less frequently of dark or thin ice. The total impact of object misclassifications is accounted for in the error analyses described in Wright and Polashenski 2018.

>This dataset is also provided in a reprojected format that does account for aircraft pitch and roll. In this work we are assessing relative fractions and not absolute areas - the difference in calculated surface fraction between images in the corrected vs raw datasets is small. Part of the manual filtering process described in the methods section includes removing those images that were not taken at or near the nominal survey altitude.

>The algorithm adjustments were tested against the same test set as used in Wright and Polashenski, 2018, and were found to not alter the overall performance.

Designation of ice type The authors state on L203 that the flight on July 25th 2017 covers first year sea ice. This does not seem justifiable for two reasons. a) the authors

provide their own definition of a FYI flight (Line 197, that 90 % of the images in the flight are FYI). Given this definition, and visual inspection of the DMS imagery from the flight, it is not obvious that the flight is over predominantly FYI. A larger percentage of images with pressure ridges and rubbled ice, indicating a long deformation history, and thus, MYI. Many images resemble the MYI depicted in Figure 11a-c and described as "common examples of ponded multiyear ice floes with characteristically blue ponds that are well consolidated by surface topography" (Line 258). b) the location of the flight line north of Ellesmere Island in the Central Arctic is over sea ice known to be the oldest and thickest ice in the Arctic, and highly unlikely to be predominantly FYI in origin. The 2017 Arctic Report Card found that the ice in this region in March is predominantly MYI (Figure 3c, Perovich et al., 2017). Given that it is well know that the ice in this region is some of the thickest ice in the Arctic (e.g. Figure 2b, Sallila et al. 2019), this area is highly unlikely to be predominantly FYI. For reference: Perovich, D., Meier, W., Tschudi, M., Farrell, S., Hendricks, S., Gerland, S., Haas, C., Krumpen, T., Polashenski, C., Ricker, R., & Webster, M. (2017). Sea Ice [in Arctic Report Card 2017], http://arctic.noaa.gov/Report-Card Sallila, H., Farrell, S. L., McCurry, J., & Rinne, E. (2019). Assessment of contemporary satellite sea ice thickness products for Arctic sea ice. Cryosphere, 13(4).

>According to the sea ice age dataset (hosted at NSIDC; citation below) there are pockets of first year ice on/around July 25th 2017 in the location of this flight line. We agree that this area is typically filled by thicker multiyear ice, but that does not exclude the possibility of there being first year ice. Visual inspections of the DMS imagery show characteristics we would expect from younger, thinner ice: darker melt ponds, dark melting ice, and less surface topography.

>Tschudi, M., W. N. Meier, J. S. Stewart, C. Fowler, and J. Maslanik. 2019. EASE-Grid Sea Ice Age, Version 4. Boulder, Colorado USA. NASA National Snow and Ice Data Center Distributed Active Archive Center. doi: https://doi.org/10.5067/UTAV7490FEPB.

Forcing conditions affecting sea ice floes in survey area L212. "To investigate melt

pond statistics across ice that experienced similar forcing conditions, two flights that contained both FY and MY ice were selected for further analysis" How do the authors know this ice has experienced similar forcing conditions throughout its lifetime? Considering the Beaufort Gyre is known to be an especially dynamic area, the ice observed during the flight surveys may have come from different regions. The ice in this region may, at the time of the survey, be experiencing uniform forcing conditions, but the assumption that all ice covered in a survey has experienced similar forcing conditions throughout its lifetime is invalid.

>By nature, MYI cannot experience the same forcing conditions as FYI over its complete lifetime. Here we are just referring to the current melt season, where the assumption that ice in a similar location on the same date experiences similar atmospheric conditions. We have added more qualifications to this description in the text.

Melt pond fraction calculation clarification: L175. How is melt pond fraction calculated? If the OSSP algorithm classifies melt ponds and submerged ice in the same category, is submerged ice included in the melt pond fraction calculation? How does the inclusion of submerged ice impact the melt pond fraction parameter?

>Melt pond fraction is calculated as: Pond area / (ice are + pond area), and we have added this information to the text. Submerged ice is included in this metric. Including submerged ice as part of melt ponds is discussed in detail in the original OSSP method document. Submerged ice is radiatively similar to melt ponds and is therefore part of the same category, and not considered a misclassification.

L177. Why do the authors choose images with open water area < 70% as a threshold for displaying melt pond fraction results? Do you include images with open water area > 70% in melt pond fraction results (Section 3.2 and 3.3).

>A single IceBridge image typically only covers 600x400m. If 70% of this is ocean, then melt pond fractions calculated from this small area are very easily skewed by large ponds (this area is well below the "aggregate scale"). Note that the images are

still processed, but the pond fraction is not shown in this plot. Even full images have a small enough area for the melt pond fraction to be skewed by large melt ponds, as shown by the orange dots in Figure 5.

Minor questions needing clarification in the text: L80 Are data collected on 15 July 2016 analyzed? This flight survey is plotted in Figure 1, but no results are shown (Figure 4).

>Yes – thank you for pointing this out. That flight was somehow missed when creating Figure 4.

L180. The authors state that data from the 20 July 2016 flight were not processed because "not enough usable images" How do the authors determine what was enough?

>Of the 1587 image frames taken on July 20, less than 30 are completely haze free. We have added these numbers to the text.

L184. Does Figure 5 follow Figure 4, and only show melt pond fraction for images with open water area < 70%?

>No – but if you look at July 24th , 2017 on Figure 4 you will see that few images in this flight were flagged as having >70% open water.

L203: Can you distinguish between the 25July2017 flight A and flight B within the text and/or in Figure 4 (where they are currently shown in the same color)?

>Yes, we have separated these flights to different colors.

L323. How is a melt pond defined in this study? Is a melt pond still a melt pond when it has melted through the sea ice? What about other features: melting snow, thaw holes, algae on ice?

>We use the definition presented in Wright and Polashenski, 2018: "Melt Ponds and Submerged Ice (MPS): applied to surfaces where a liquid water layer completely submerges the ice."
>A melt pond is no longer a melt pond when it has melted through the ice. Melting snow falls into the ice/snow category, algae and sediment laden ice are not defined but would likely be assigned to the dark ice category depending on their color and brightness.

Figure 4. Bottom figure. For the 17 July 2017 and 18 July 2017 flights, it looks like there are no images remaining for analysis. Is that correct? Can you provide the total number of images analyzed for each flight, and total discarded? Perhaps this information could be included in a table or added to the figure.

>We have added this information in the methods section.